# The Vital Role of Melatonin and Its Metabolites in the Neuroprotection and Retardation of Brain Aging

**DOI:** 10.3390/ijms25105122

**Published:** 2024-05-08

**Authors:** Georgeta Bocheva, Dimitar Bakalov, Petar Iliev, Radka Tafradjiiska-Hadjiolova

**Affiliations:** 1Department of Pharmacology and Toxicology, Medical University of Sofia, 1431 Sofia, Bulgaria; 2Department of Physiology and Pathophysiology, Medical University of Sofia, 1431 Sofia, Bulgaria

**Keywords:** melatonin, brain aging, neuroprotection, neurodegenerative disorders

## Abstract

While primarily produced in the pineal gland, melatonin’s influence goes beyond its well-known role in regulating sleep, nighttime metabolism, and circadian rhythms, in the field of chronobiology. A plethora of new data demonstrates melatonin to be a very powerful molecule, being a potent ROS/RNS scavenger with anti-inflammatory, immunoregulatory, and oncostatic properties. Melatonin and its metabolites exert multiple beneficial effects in cutaneous and systemic aging. This review is focused on the neuroprotective role of melatonin during aging. Melatonin has an anti-aging capacity, retarding the rate of healthy brain aging and the development of age-related neurodegenerative diseases, such as Alzheimer’s disease, Parkinson’s disease, Huntington’s disease, multiple sclerosis, amyotrophic lateral sclerosis, etc. Melatonin, as well as its metabolites, N1-acetyl-N2-formyl-5-methoxykynuramine (AFMK) and N1-acetyl-5-methoxykynuramine (AMK), can reduce oxidative brain damage by shielding mitochondria from dysfunction during the aging process. Melatonin could also be implicated in the treatment of neurodegenerative conditions, by modifying their characteristic low-grade neuroinflammation. It can either prevent the initiation of inflammatory responses or attenuate the ongoing inflammation. Drawing on the current knowledge, this review discusses the potential benefits of melatonin supplementation in preventing and managing cognitive impairment and neurodegenerative diseases.

## 1. Introduction

Melatonin (N-acetyl-5-methoxytryptamine) is a multifunctional hormone with a universal distribution in nature [1,2] and remarkable essential functions [3,4]. It can be produced in all living species including plants [5,6,7]. Melatonin was first identified in the bovine pineal glands by the dermatologist Aaron Lerner et al. in 1958 [8]. It was thought that melatonin could be uniquely released by the pineal gland, playing a major role in the regulation of circadian rhythms and seasonal biorhythms [9]. Pineal-released melatonin can be measured at lower levels in the blood than in the cerebrospinal fluid (CSF) of the third ventricle, suggesting its neuroprotective role against oxidative stress [10]. To this date, many extra-pineal sites of melatonin synthesis have been established, like bone marrow, retina, lens, cochlea, lungs, liver, kidney, pancreas, thyroid gland, female reproductive organs, and skin [11,12,13,14,15,16,17,18]. Melatonin exhibits a diverse protective potential that includes a high capacity to mitigate oxidative stress, protect mitochondrial functions, modulate the immune system, reduce inflammation, modify the cell response and cytokine release, and enhance circadian rhythm amplitudes. These different mechanisms of melatonin’s action contribute to its neuroprotective effects, potentially retarding the process of aging [19,20] and the onset of age-related neurodegenerative diseases, such as Alzheimer’s disease [21], Parkinson’s disease [22], Huntington’s disease, multiple sclerosis, amyotrophic lateral sclerosis (ALS), etc. [23].

Neurodegenerative diseases are a heterogeneous group of aging-related disorders, characterized by a significantly increased neuronal cell death and a progressive loss of function. They are often linked to pathological protein formation, as well as aggregate formation in many patients, causing cytotoxicity. The cytotoxic events include mitochondrial dysfunction, oxidative stress, DNA damage, impaired protein degradation systems, synaptic dysfunction, and cell cycle re-entry [24]. Since mitochondrial dysfunction with a reduced capacity for the production of adenosine triphosphate (ATP) has been implicated in the pathogenesis of brain age-associated decline and major neurodegenerative diseases [25,26], melatonin could be an excellent novel therapeutic option. As a highly powerful antioxidant, melatonin, along with some of its metabolites, is capable of preserving mitochondrial bioenergetic function in the brain by reducing oxidative damage; thus, it may delay the progression of the aging phenotype and the development of age-related neurodegenerative diseases [27,28]. Furthermore, melatonin could be implicated in the treatment of neurodegenerative conditions, by modifying their characteristic low-grade neuroinflammation. It can either prevent the initiation of inflammatory responses or attenuate the ongoing inflammation [29].

## 2. Melatonin and Its Metabolites as Determinants in Oxidative Stress

Melatonin is the oldest antioxidant with the ability to reduce oxidative damage through non-receptor-mediated mechanisms, by scavenging the toxic radicals directly (both ROS and RNS) [30,31,32], and via the inhibition of their generation, especially at the mitochondrial level [27]. It can be found in sufficiently high concentration in mitochondria, either due to rapid uptake [33] or due to its synthesis on site [34]. Melatonin, with its positional advantage, is able to immediately scavenge the toxic free radicals formed in abundance in mitochondria [35,36], as well as to reduce the formation of the superoxide anion radical (O_2_•^−^), a process referred to as radical avoidance [37]. The advantage of melatonin over classical antioxidants is that its final kynuric metabolites, N1-acetyl-N2-formyl-5-methoxykynuramine (AFMK) and N1-acetyl-5-methoxykynuramine (AMK), share the ability to scavenge and detoxify reactive oxygen and nitrogen species, creating a cascade of antioxidant reactions [38,39]. 

Additionally, melatonin demonstrates highly effective antioxidant properties, through its capacity to stimulate the production of antioxidant enzymes [40], Sirtuin (SIRT) 3, etc., via indirect melatonin receptors (MT1/MT2)-mediated action [41]. Indeed, through the activation of MT1/MT2, melatonin up-regulates the expression of antioxidant genes [42]. The antioxidant enzymes, e.g., superoxide dismutase (SOD), catalase (CAT), and glutathione peroxidase (GPx), are able to degrade the weakly reactive ROS [43,44]. But the most reactive and destructive species, like peroxynitrite (ONOO^−^) and the hydroxyl radical (•OH), are not degraded by the enzymes mentioned above. They can only be removed by direct highly efficient scavengers like melatonin, AFMK, and AMK, thereby limiting the ROS-generated oxidative damage on different cell structures and molecules including lipids, b-oxidation cycle compounds, or mitochondrial DNA (mtDNA) [30].

Recently, some SIRTs (class of nicotinamide adenine dinucleotide (NAD^+^)-dependent deacetylases) demonstrated the epigenetic ability to deacetylate histone and non-histone targets, thus modulating the expression of the genes implicated in oxidative stress response and apoptosis [45]. Therefore, SIRTs could play crucial roles in the regulation of brain function during the ageing process.

## 3. Melatonin in Aging

Biological aging is a natural phenomenon characterized by a progressive loss of functional capacity, physiological integrity, and morphological features of the human body. Mechanisms underlying the aging process include oxidative stress, mitochondrial dysfunction, inflammaging, disruption of circadian rhythms, proteostasis, telomere attrition, genomic instability, epigenetic alterations, and decreased capacity of tissue repair [46,47]. Circadian clocks are vital for the regulation of the human health, by orchestrating physiological and neuroendocrine functions. Chronobiological aging is associated with a decrease in melatonin secretion, a decline in circadian rhythm, and a dampening of circadian gene expression [48] that can augment oxidative damage through an increased generation and accumulation of ROS and RNS [49]. The “free radical theory of aging” has been the focus of many scientists for over 50 years [50]. During the process of aging, ROS and RNS are produced through cellular oxidative metabolism. At the subcellular level, mitochondria are the major source of generation of •OH and ONOO^−^ [51]. The excessive production of these destructive species, causes enhanced mitochondrial oxidative stress as well as mtDNA mutations, and occurs along with human aging and age-related pathologies [52,53,54]. 

Accumulating evidence supports a strong connection between the process of aging and the decline in mitochondrial quality and function. The aging of mitochondria is characterized by a significant increase in ROS generation, a decrease in antioxidant defense, and a reduction in oxidative phosphorylation and ATP production. Disturbances in the mitochondrial redox balance further drive cellular senescence. This age-linked mitochondrial impairment enhances mitochondria-mediated apoptosis, thus leading to an increase in the number of apoptotic cells. In addition, mtDNA with functional decline, as a consequence of aging, results in the further enhancement of ROS production [55]. Lately, it has been thought that most mtDNA mutations are caused by replication errors made by the mtDNA polymerase [56]. During the process of aging, such defects in mtDNA replication, along with the failure of their repair mechanisms might lead to an accumulation of mutations which could further increase mitochondrial impairment and the augmentation of oxidative damage. Therefore, mitochondrial dysfunction determines the rate of aging [54,57]. 

Some intracellular enzymes outside the mitochondria (e.g., xanthine oxidase, monoamine oxidase, and NADPH oxidases) also impact ROS production, with advancing age [58,59,60]. 

Experimental studies on young rats with surgical pinealectomy showed accelerated oxidative damage in many tissues, as well as premature and faster aging [61], highlighting the vital role of the indolic hormone melatonin in the process of aging.

Melatonin is a mitochondria-targeted comprehensive antioxidant that is able to reduce the mitochondrial production of free radicals or to detoxify them, as well as indirectly activating the mitochondria-located SOD2, which may lead to the retardation of the rate of skin [62,63,64] and systemic aging [27,65,66]. Melatonin’s concentration is found at higher levels in mitochondria than in other cellular organelles, suggesting its significant role in mitochondrial processes [67]. The multiple beneficial protective actions of this indolic hormone, at the mitochondrial level, are well documented [68]. With the stimulation of SIRT3, localized in mitochondria, melatonin gives rise to the deacetylation and stimulation of SOD2. The activation of the SIRT3/SOD2 antioxidant signaling pathway due to melatonin limits the mitochondrial oxidative damage and cytochrome C release, thus reducing mitochondria-related apoptosis [41,69]. 

Indeed, independent of receptors, melatonin works as a mitochondrial protector, maintaining optimal mitochondrial physiology [27,70] not only by quenching ROS, but also by inhibiting the mitochondrial permeability transition pore (MPTP) [71], activating uncoupling proteins, regulating mitochondrial biogenesis and dynamics [37], optimizing respiratory chain complexes, enhancing mitochondrial ATP production, and regulating the process of mitophagy (removal of damaged mitochondria) [67,72].

During the process of aging, increasing levels of inflammatory cytokines are frequently measured. The imbalance between inflammatory and anti-inflammatory mechanisms in the aging phenotype causes a low-grade chronic inflammation, known as the “inflammaging” state [73]. “Inflammaging” is caused by both chronic antigen stimulation over the human lifespan and also a continued exposure to oxidative stress. These factors contribute to a remodeling of the immune system, with changes in cellular and humoral immunity potentially leading to a shift of the immune system to an inflammatory mode, in advanced age. In fact, immunity appears to play a significant role in the process of aging and age-related diseases [74]. Overall, melatonin may act like either a pro- or anti-inflammatory molecule in different conditions [29,75]. During the aging process, melatonin acts preferentially as an anti-inflammatory agent on aging-related, low-grade inflammation [76]. Melatonin stimulates SIRT1 and their anti-inflammatory activities overlap during the process of aging [77]. SIRT1 functions as an epigenetic regulator of aging that alleviates inflammation by down-regulating Toll-like receptor (TLR)-4, which mediates pro-oxidant effects through the NF-κβ signaling pathway [29]. Melatonin, via the inhibition of either TLR-4 or toll-receptor-associated activator of interferon (TRIF), can suppress the release of several pro-inflammatory cytokines like TNFα, IL-1β, IL-6, and IL-8 [78,79].

Summarizing, melatonin, with its capacity to mitigate oxidative stress, to protect mitochondrial functions, to modulate the immune system, to ameliorate inflammation, to enhance circadian rhythm amplitudes, and to exhibit neuroprotection, beneficially results in the retardation of aging. 

## 4. Brain Aging Defense of Melatonin

The brain is the most complex human organ, built from two main cell types—neurons and glia. They use different metabolic pathways to produce energy. As the center of the nervous system, the brain orchestrates many processes required for body homeostasis. It has an important role in physiology and metabolism; therefore, it may also be at the center of aging. Brain aging is an irreversible process and its progression is a consequence of the interactions between genetic and environmental factors (infections, trauma, drugs or neurotoxin exposure, etc.). In the aging brain, pathological changes at both the cellular and tissue level can cause cognitive and motor impairment, memory loss, and other phenotype characteristics. 

Microglia represent macrophage-like cells, which are the resident immune cells of the brain. During the process of aging, microglial cells increase in both number and size and undergo morphological changes. Moreover, aging induces a shift between pro-inflammatory and potentially cytotoxic (M1), as well as anti-inflammatory and regenerative (M2), microglial phenotypes [80]. Microglial activation due to aging can lead to neuroinflammation and neurodegeneration. Aberrant activated microglia generate an excessive ROS/RNS production that triggers the NF-κB signaling cascade and neuroinflammation, promoting neuronal damage and cell death [81]. During the process of aging, there is an accumulation of mtDNA oxidative damage of microglia [82]. Additionally, neuronal cells demand high amounts of energy in order to function and are highly sensitive to any mitochondrial dysfunction and oxidative phosphorylation system (OXPHOS) defects [83]. Due to metabolic differences, neuronal and glial mitochondria produce different amounts of ROS/RNS. Although they produce less mitochondrial ROS (mtROS), neurons are more vulnerable than glia to oxidative damage, because they have fewer antioxidants [84]. In fact, neurons are rarely replaced during the human lifespan. Some areas of the brain like the hippocampus, substantia nigra, amygdala, and frontal cortex are more sensitive to oxidative stress [85]. Spontaneous errors in the mtDNA replication machinery and failure in mtDNA repair might cause the accumulation of mutations, resulting in mitochondrial dysfunction and the bioenergetic failure of the neural cell. mtROS initiate and maintain a permanent cell-cycle arrest, known as cellular senescence [53,86]. Therefore, mitochondria are a key factor in the development of pro-aging characteristics of brain cells’ senescence [87,88,89]. Studies have shown that senescent microglia exhibit increased mitochondrial DNA damage [82], a reduced ability to clear debris (reduced phagocytosis), and increased secretion of molecules that promote inflammation (pro-inflammatory cytokines) [90,91]. These changes can further contribute to the spread of senescence to surrounding cells. The prevention of microglial activation and the normalization of mitochondrial function in aged individuals represent potential therapeutic strategies, which guarantee a healthy brain aging process.

Brain aging also leads to the altered function of the main biological clock, the hypothalamic suprachiasmatic nucleus (SCN), resulting in a decrease in melatonin synthesis in the pineal gland and circadian rhythm disruption [92]. The disruption of the circadian rhythm, in turn, leads to various physiological changes, such as disturbances in the sleep–wake cycle, metabolic abnormalities, neurodegeneration, oxidative stress, etc. [93]. The age-related decline of melatonin release impairs mitochondrial homeostasis in neurons, thus playing a major role in the emergence of neurological abnormalities and accelerated aging. Exogenous melatonin has been proposed not only as a potential circadian synchronizer [94], but also as a regulator of redox homeostasis in the brain during aging [95].

Brain mitochondrial functional changes within aging are thought to be the major contributor to the aging process. The clearance of malfunctioning or damaged mitochondria is essential for controlling brain cells’ homeostasis. A recent study demonstrates a new way in which melatonin combats brain inflammation. This novel mechanism involves the activation of mitophagy that selectively removes damaged mitochondria. By eliminating these damaged structures, melatonin helps to maintain a healthy brain environment [96]. 

Melatonin also controls microglial activation by inhibiting the production of pro-inflammatory cytokines and chemokines both in vitro and in vivo [97,98,99,100]. It is able to inhibit TLR-4 expression and caspase-3 activation in hypoxic BV-2 microglial cells [100]. Melatonin has been shown to effectively suppress IL-1β and TNF-α at both the mRNA and protein levels, as well as to down-regulate the inducible nitric oxide synthase (iNOS) in activated BV2 microglial cells [101]. In fact, iNOS is a marker of the M1 microglia phenotype, but not of the M2 type [80]. It was suggested that melatonin may promote the M2 polarization of microglia and suppress pro-inflammatory responses in an injured spinal cord, facilitating functional recovery [102]. Also, the administration of melatonin can lead to the decreased expression of the NLRP3 (nucleotide-binding domain leucine-rich repeat and pyrin domain containing receptor 3) inflammasome in rats with spinal cord injury, thus exerting neuroprotective effects on motor neurons [103]. Furthermore, melatonin is able to inhibit NLRP3 activation and to reduce pro-inflammatory factors, via promoting the activation of Nrf2/ARE signaling [104]. The NLRP3 inflammasome is a protein complex that initiates an inflammatory form of cell death and triggers the release of IL-1β and IL-18 and has also been implicated in neurodegenerative diseases [105,106]. Therefore, the suppression of the NLRP3 inflammasome by melatonin controls neuroinflammation and attenuates mitochondrial dysfunction. 

Another study evaluated the effect of melatonin and its final metabolites, AMK and AFMK, on neuronal NOS (nNOS) both in vitro and in rat striatum in vivo. It was found that only melatonin and AMK, but not AFMK, inhibit nNOS in a dose–response manner. In vivo, the potency of AMK to inhibit rat striatal nNOS activity was higher than that of melatonin [107]. Additionally, melatonin and AMK have been reported to enhance cognitive processes. Melatonin crosses the blood–brain barrier and immediately converts to AMK, in brain tissue. Endogenous or exogenous AMK enhance long-term object recognition memory in aging mice, suggesting a therapeutic potential of AMK to improve or retard memory decline [108,109]. Based on these findings, melatonin and kynuramine AMK possess the power to decrease the low-grade inflammation and alleviate neuroinflammation in the aging brain. 

## 5. Melatonin in Neuroprotection

### 5.1. Pathogenesis of Neurodegeneration

Brain aging and neurodegeneration very often overlap. Advancing age is a major risk factor for most neurodegenerative disorders, which occur prevalently in aged individuals [110]. Signs of brain aging have been observed in both patients and animal models with Alzheimer’s [111,112] and Parkinson’s disease [113], because aged brains become highly susceptible to neurodegeneration [114,115]. Redox dysregulation [116], mitochondrial dysfunction [25,26,117,118], and cellular senescence [115,119] are the key modulators implicated in many neurodegenerative diseases. Additionally, a growing body of research indicates that these different age-related neurodegenerative conditions share a common inflammatory mechanism with the activation of the NLRP3 inflammasome complex in microglia and peripheral monocytes, with a consequent increased production of pro-inflammatory cytokines (e.g., IL-1β and IL-18) [120]. The overactivation of NLRP3 is linked to mitochondrial damage and abnormal mitophagy.

Further, in numerous neurological diseases, amyloid deposits composed of α-synuclein protein, microtubule-associated protein tau, and amyloid beta (Aβ) peptide are found. These deposits have a different nature and localization in the brain, which helps in the sub-classification of the neurodegenerative disorders. An abnormal extracellular Aβ deposition due to the suppression of Aβ clearance from damaged astrocytes during aging is characteristic for Alzheimer’s disease [121]. Abnormal intracellular inclusions containing hyperphosphorylated and aggregated tau protein (p-tau), an integral component of the filaments of neurofibrillary tangles, are additionally found in the same disorder [122]. Deposits of Aβ and p-tau are pathological hallmarks in patients with Alzheimer’s, which positively correlates with progressive cognitive decline [123]. The presence of aggregated and phosphorylated α-synuclein protein (Lewy bodies) in the subcortical regions of the brain is a pathological characteristic for Parkinson’s disease [124], but it is not only restricted to this disorder [125]. The α-synuclein pathology is a feature of other synucleinopathies with progressive dementia, such as Parkinson’s disease with dementia (PDD) and dementia with Lewy bodies (DLBs). It also frequently occurs in Alzheimer‘s disease, where α-synuclein contributes to secondary symptoms. Moreover, the concomitance of α-synuclein and tau pathology is not rare [126]. Both are often found in neurons of the amygdala in DLBs patients [127]. α-synuclein was initially demonstrated to bind to tau and to interfere with its normal interaction with tubulin, thereby interrupting tau’s physiological functions [128]. Furthermore, α-synuclein may recruit kinases that promote tau phosphorylation, potentially leading to the formation of tau/α-synuclein co-oligomers [129,130,131].

Large amounts of oxidative stress are thought to be a major contributing factor in most neurodegenerative diseases, playing a role similar to its involvement in the process of brain aging [116,132,133,134,135,136]. Oxidized proteins and lipid peroxidation were observed in patients with mild cognitive impairment and early stage Alzheimer’s disease, which possibly precede Aβ accumulation [137]. Aβ peptides also exhibit pro-oxidant and proinflammatory properties. The most toxic ones are Aβ monomers and oligomers. For example, a synergic effect between the amyloid β1-42 oligomer and oxidative stress was observed in the development of the Alzheimer’s disease-like neurodegeneration of hippocampal cells [138]. In vivo studies confirmed this observation. By using the single intracerebroventricular (icv) injection of protofibrillar Aβ1-42 in the hippocampus of rats, amyloid deposits have led to both an increase in ROS production and enhanced lipid peroxidation, as well as the inhibition of antioxidant enzyme activity in the hippocampus, cortex, and striatum regions of the brain, along with impaired long-term memory and anxiety-like behavior [139]. An elevated oxidative stress near Aβ deposits predisposes to neurotoxicity and neuronal loss. Further, oxidative stress is aggravated by mitochondrial dysfunction and endoplasmic reticulum stress (Figure 1). Conversely, Aβ peptides induce the lipid peroxidation of brain cell membranes that, in turn, trigger Aβ production, by increasing the activity of β-secretase (BACE1) and γ-secretase. The presence of both β-amyloid and tau deposits drives early Alzheimer’s disease decline [140]. 

Additionally, the release of pro-inflammatory mediators due to Aβ peptides and oligomers are seen in microglia [141] and neurons [142]. Since Aβ peptides are present in the cerebrospinal fluid (CSF) of healthy individuals, their efficient removal, especially during sleep, is critically important [143]. Therefore, sleep disturbances, which appear to be a common finding in many neurodegenerative disorders, impair Aβ clearance [144,145].

Mitochondrial dysfunction is a common feature also in Parkinson’s disease because aggregated α-synuclein preferentially binds to mitochondria, and its interaction with mitochondrial electron transport chain (ETC) increases the production of mitochondrial ROS, stimulating mtDNA damage. This interaction also results in decreased both respiratory capacity and ATP production. Mitochondrial aggregation of α-synuclein promotes MPTP opening, calcium diffusion, cytochrome C release, and mitochondrial swelling, which ultimately leads to apoptosis [146]. It has been found that α-synuclein interaction with fusion proteins (e.g., OPA1, MFN-1, and MFN-2) is associated with enhanced mitochondrial fragmentation and bioenergetic alterations in induced pluripotent stem cell (iPSC)-derived dopaminergic neurons [147]. Furthermore, pathological α-synuclein and reactive microglia can potentiate each other, causing loss of dopaminergic neurons and accelerated neurodegeneration in Parkinson’s disease [148,149]. Additionally, activated microglia facilitate the transfer of α-synuclein, including via exosomal pathways, contributing to the progression of α-synuclein pathology [150,151,152].

### 5.2. Neuroprotection of Melatonin

Numerous experimental studies and clinical trials confirm the therapeutic neuroprotective potential of melatonin and its derivatives. They exert powerful neuroprotection through myriad of different mechanisms, allowing for the prevention of neurodegeneration and/or cognitive improvement, along with sleep maintenance [153,154,155,156,157,158]. Studies have demonstrated that melatonin inhibits the nuclear translocation of NF-κBp65 and the activation of glycogen synthase kinase (GSK)-3β, through melatonin receptor activation in Aβ1-42-treated SH-SY5Y neuroblastoma cells [159]. GSK3β serves as a crucial link between the beta-amyloid and τ-tangle pathologies. It regulates Aβ production, possibly by interfering with β-amyloid precursor protein (APP)-cleaving secretases [160]. By blocking the GSK3β pathway from melatonin, a partial decrease in Aβ1-42-induced elevation in BACE1 has also been seen [159]. Several in vivo studies demonstrate that melatonin is able to prevent neurodegeneration and cognitive deficits in Aβ1-42-induced neurotoxicity in the hippocampus of mice [161,162]. Aβ1-42, injected icv, would cause synaptic dysfunction, memory impairment, and hyperphosphorylation of the tau protein. The intraperitoneal (i.p) administration of melatonin (10 mg/kg/24 h) for 3 weeks has attenuated the impairment of memory and has decreased tau hyperphosphorylation, through PI3K/Akt/GSK3β signaling in Aβ1-42-treated mice. In addition, melatonin has decreased apoptosis in the same model, by suppressing the overexpression of caspase-3, caspase-9, and PARP-1 [163], as well as demonstrating a significant increase in the Bcl-2/β-actin and PP2A/β-actin proteins [160]. Similarly, early melatonin administration in transgenic mice significantly decreased up-regulated apoptotic-related caspase-3 and Bax, as well as decreasing the level of thiobarbituric acid-reactive substances (TBRASs) in the brain [164]. It has also been found that the supplementation of melatonin reduces the number of apoptotic neurons and increases the choline acetyltransferase activity in the frontal cortex and hippocampus of an APP 695 transgenic mouse model of Alzheimer’s disease [165]. 

Exogenous melatonin administration (50 mg/kg, i.p. for 40 days) was able to mitigate the cognitive decline in pinealectomized (pin) adult Sprague Dawley rats, Aβ1-42 icv injected rats, and a combination of both a pin and Aβ1-42 (pin+Aβ1-42)-treated rat model, whereas it could correct the elevated anxiety only in the pin+Aβ1-42 model [166]. As a result, long-term melatonin administration alleviates behavioral and cognitive deficits and reduces Aβ aggregation and deposition, probably through promoting Aβ clearance via glymphatic drainage [167,168,169,170,171]. Of note, about 95% of all patients with Alzheimer’s disease have the sporadic, late-onset form, which demonstrate only reduced Aβ clearance [172]. Moreover, by using the molecule receptor saturation binding assay, melatonin is established to bind with high specificity and affinity to β-amyloid [173]. Therefore, melatonin plays a crucial role in protecting the brain from Aβ-induced neurotoxicity. Unfortunately, there is a lack of convincing clinical evidence that suggests melatonin affects Aβ pathology. Most trials in humans with Alzheimer’s disease focus on sleep quality and cognitive performance after melatonin treatment. 

Studies have shown plasma antioxidants enzyme activities are significantly decreased in patients with mild cognitive impairment, which may be a prodromal stage of neurodegenerative diseases [174]. Moreover, the decline in cognitive function positively correlates with the depletion of antioxidant defense [175]. Therefore, preventing and treating brain oxidative damage, one of the earliest pathophysiological processes, necessitates an increased antioxidant intake [176]. The antioxidant activities of the widely known hormone melatonin and its brain metabolite AMK are well studied and documented in a lot of experimental research papers on the topic of neurodegeneration. As a mitochondrially targeted robust antioxidant and scavenger of toxic free radicals, melatonin mitigates oxidative stress in the brain [164,166,177,178,179,180,181,182]. Decreased oxidative stress markers were also measured in patients with Parkinson’s disease [183] and amyotrophic lateral sclerosis [184], after melatonin administration. In addition, in the same placebo-controlled clinical trial, patients with Parkinson’s disease treated with 25 mg melatonin for 12 weeks showed a significant increase in mitochondrial complex I enzymatic activity, although membrane fluidity was unaltered [183]. Another recent randomized clinical study on patients with Parkinson’s evaluated the impact of melatonin supplementation (10 mg/24 h for 12 weeks) on total antioxidant capacity (TAC) and total glutathione (GSH) levels. Melatonin supplementation, in this study, resulted in a concomitant elevation of TAC and GSH, alongside improvements in clinical and metabolic parameters [185].

Lately, many research data suggest a close relationship between oxidative stress and impaired mitophagy function in the pathogenesis of major neurodegenerative diseases [186]. Transcription factor EB (TFEB) promotes mitophagy, by regulating the autophagosome–lysosomal fusion and autophagosome formation [187]. TFEB-mediated mitophagy supports the clearance of damaged mitochondria and the removal of excessive toxic radicals. A recent study showed that melatonin, administrated for 3 months through drinking water, was able to induce TFEB nuclear translocation, promote mitophagy, and increase mitophagy-related protein PINK1 in the brains of APP/PS1-transgenic mice. Moreover, melatonin inhibited NLRP3 inflammasome activation, decreased ROS levels, and improved cognitive function in the same mouse model [188].

## 6. Pharmacokinetics of Exogenous Melatonin and Clinical Administration

Oral formulations of melatonin show variability in its absorption and metabolism. The bioavailability of orally supplemented melatonin ranges from 2.5% to 33% and the food delays its absorption [189,190]. After oral administration melatonin undergoes extensive first-pass hepatic metabolism that is regulated by cytochrome P450 oxidases [191]. Exogenous, similar to endogenous melatonin, is metabolized mainly in the human liver but also in the brain, skin, and lungs through various CYP1 isozymes, mostly CYP1A2 [62,192]. CYP1A2 shows higher activity in men that could cause gender differences after melatonin supplementation [193]. Melatonin is degraded to 6-hydroxymelatonin and N-acetylserotonin which are excreted in urine. 

Melatonin is available as immediate, prolonged-release or a combined formulation with a great variability in the doses. Immediate-release melatonin has a short half-life (up to 1 h). Prolonged-release melatonin has longer half-life (about 4 h) and exhibits more physiologic release in comparison to the pharmacologic delivery of immediate-release formulations of melatonin. Both galenic forms are well tolerated [194]. Exogenous melatonin has clinically proven beneficial chronobiotic effect, as well as its effectiveness in primary insomnia [195,196]. Moreover, most data for efficacy and safety of long-term administration of melatonin are based on its use in primary insomnia. It should be noted that exogenous melatonin has a high safety profile and no rebound or suppression of endogenous melatonin production following discontinuation are seen [197,198].

A possible link between cognitive decline and poor sleep quality has been suggested. Therefore several clinical trials of melatonin supplementation were performed in neurodegenerative diseases [199]. In a placebo-controlled multicenter trial, melatonin administrated for 6 months in 2 mg prolonged-release tablets demonstrated a significantly better cognitive performance and sleep efficiency in patients with mild to moderate Alzheimer’s disease [153]. Another trial with melatonin was performed in Parkinson’s disease patients with poor sleep quality. After 4 weeks, the supplementation of 2 mg prolonged-release melatonin showed beneficial effects on sleep disruption together with improved non-motor symptoms and quality of life of these patients [200]. On the other hand, Gilat M et al. did not find an effect of prolonged-release 4 mg melatonin for 8 weeks on rapid eye movement sleep behavior of Parkinson’s disease patients [201]. 

## 7. Discussion and Conclusions

The goal of this review is to summarize the action of melatonin and its effects on aging and degeneration of the brain. 

Melatonin’s neuroprotection in various in vitro and animal models of neurodegeneration could be explained by its potential antioxidant, anti-inflammatory, anti-Aβ aggregation properties, by regulation of apoptosis, and protection of the cholinergic system (Figure 2). 

Numerous clinical trials also confirm melatonin’s beneficial effects in the retardation of brain aging and in the neuroprotection of progressive neurodegenerative disorders, which are frequently found with insomnia comorbidity. For optimal neuroprotective effects of melatonin, early supplementation is crucial. 

However, more targeted studies of melatonin supplementation in aged individuals with signs of brain aging and neurodegeneration are needed and must be performed in the future. This need for high-quality clinical trials is necessary for a comprehensive research and for an evaluation of the effect of supplemented melatonin in different doses and formulations to design the optimal effective treatment for retardation of neurodegeneration.

## Data Availability

All data appear in the manuscript. For further inquiries, please contact the first author or corresponding author.

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
