# Peer review of "The Vital Role of Melatonin and Its Metabolites in the Neuroprotection and Retardation of Brain Aging"

_ijms, 2024, doi:10.3390/ijms25105122_

Round 1
Reviewer 1 Report
Comments and Suggestions for Authors
The review is written on a fairly popular topic and it's unclear what makes it different. Authors should choose a new focus or aspect and emphasise on it, clearly writing how their review differs from similar ones. Perhaps add a section with general information on dosages and pharmacokinetics of melatonin, linking the doses used in studies for antioxidant effects with those actually used and safe for patients. The review is not structured enough, it is better to table the main lines of action of melatonin with specific targets and references to the works in which this has been shown. In addition, the Conclusion section should be highlighted and the findings and conclusions that can be drawn from the available data at this time should be written more clearly and in more detail.
Author Response
We greatly appreciate the reviewers’ critique that has improved final presentation of the manuscript.
Please, see our answers below for details.
The review is written on a fairly popular topic and it's unclear what makes it different. Authors should choose a new focus or aspect and emphasise on it, clearly writing how their review differs from similar ones.
Reply: We agree that there are numerous reviews about the antioxidant effects of melatonin or its anti-inflammatory networks in the brain, which we cited in our manuscript (20; 29; 133, 135, etc). Our goal was to summarize the multifaceted action of melatonin and its effects on aging and degeneration of the brain. Most reviews separate the effects of melatonin in different neurodegenerative disorders, which makes a potential duplications of its effects because they often share the same pathological mechanisms (“redox dysregulation [116], mitochondrial dysfunction [25,26,117,118] and cellular senescence [115,119]”; “these different age-related neurodegenerative conditions share a common inflammatory mechanism with activation of the NLRP3 inflammasome complex in microglia and peripheral monocytes…”). While some neurodegenerative diseases share common pathological mechanisms, we explore how melatonin's influence on these shared mechanisms may have effects on neurodegeneration as a whole. This approach provides a broader understanding compared to reviews focusing on individual disorders and that is why we trust in the uniqueness of our review.
Perhaps add a section with general information on dosages and pharmacokinetics of melatonin, linking the doses used in studies for antioxidant effects with those actually used and safe for patients.
Reply: We added the proposed section
The review is not structured enough, it is better to table the main lines of action of melatonin with specific targets and references to the works in which this has been shown.
Reply: This has been done from many authors and we cited their work (e.g. 133)
In addition, the Conclusion section should be highlighted and the findings and conclusions that can be drawn from the available data at this time should be written more clearly and in more detail.
Reply: This has been corrected
Reviewer 2 Report
Comments and Suggestions for Authors
This article addresses the links between melatonin and brain aging. Authors specifically focused on potential benefits of melatonin supplementation in preventing and managing cognitive impairment and neurodegenerative diseases.
It is a well-written and useful review of the current data, interesting and should be of great interest of the readers.
The field is relevant and reviews have been recently published regarding the relationship between melatonin and aging. To stand out from previous reviews, the manuscript should also include the melatonin treatment effects on neurodegenerative disorders (for each neurodegenerative disorder), making a clear distinction between in vitro/in vivo studies and clinical trials. Melatonin dosage on the studies should be considered and discussed. Furthermore, the topic melatonin in neurodegenerative disorders should be divided in sub-topics considering each of the neurodegenerative disorder that is mentioned (for example 5.1. AD; 5.2. PD;…). In these subtopics might be included the role of melatonin treatment as suggested previously. Is there any differences between sexes??
A conclusion and perspective section must be included, clearly showing the importance of melatonin in neurodegenerative diseases. What about future research? What can be done to maintain melatonin levels?
Minor points should be addressed:
1. In order to make the manuscript easy to understand I would suggest a graphical abstract.
2. The schemes (figures) should be improved. They are very confusing.
3. The body of the review should be also reorganized. It is crucial to create links between the discussed research findings and the research question. It will establish a more coherent review article.
4. A discussion section should be included in order to summarize and connect the important aspects of the existing literature.
Comments on the Quality of English LanguageMinor editing is recommended
Author Response
The manuscript entitled “The vital role of melatonin and its metabolites in neuroprotection and retardation of brain aging” has been revised and supplemented following reviewer’s suggestions and recommendations.
Reply: We thank the Reviewer for his/her effort to improve our presentation. We revised the manuscript as requested. To avoid duplications in the effects of melatonin in each neurodegenerative disease we wrote its main effects in the process of neurodegeneration.
A conclusion and perspective section must be included, clearly showing the importance of melatonin in neurodegenerative diseases.
What about future research? What can be done to maintain melatonin levels?
Reply: It has been included.
In order to make the manuscript easy to understand I would suggest a graphical abstract.
Reply: The topic in general is difficult and cannot be simplified by a graphical abstract.
The schemes (figures) should be improved. They are very confusing.
Reply: We have attempted to improve the Scheme 2. The red color of the arrows is used for better visualization of the multifaceted action of melatonin in neurodegeneration. However, these Schemes aim to encompass the variety of mechanisms by which melatonin offsets the negative effects of the neurodegenerative process but to try and simplify them further could cause inaccuracies or errors.
A discussion section should be included in order to summarize and connect the important aspects of the existing literature.
Reply: It has been included.
Round 2
Reviewer 1 Report
Comments and Suggestions for Authors
The manuscript can be accepted in present form